# Dental Pain Medication Prescriptions in Minas Gerais, Brazil (2011–2021): A Time-Series Analysis

**DOI:** 10.3390/ijerph20186795

**Published:** 2023-09-21

**Authors:** Alex Junio Silva Cruz, Maria Auxiliadora Parreiras Martins, Victor Santos Batista, Henrique Pereira de Aguilar Penido, Jacqueline Silva Santos, Thiago Rezende dos Santos, Woosung Sohn, Lia Silva de Castilho, Mauro Henrique Nogueira Guimarães Abreu

**Affiliations:** 1Graduate Dental Program, School of Dentistry, Universidade Federal de Minas Gerais, Avenida Antonio Carlos, Belo Horizonte 31270901, Brazil; aame0590@ufmg.br; 2Department of Pharmaceutical Products, Universidade Federal de Minas Gerais, Avenida Antonio Carlos, Belo Horizonte 31270901, Brazil; auxiliadorapmartins@ufmg.br; 3Undergraduate Dental Program, School of Dentistry, Universidade Federal de Minas Gerais, Avenida Antonio Carlos, Belo Horizonte 31270901, Brazil; victorsb13@ufmg.br; 4Undergraduate Math Program, Institute of Exact Sciences, Universidade Federal de Minas Gerais, Avenida Antonio Carlos, Belo Horizonte 31270901, Brazil; hpenido@ufmg.br; 5Oral Health Department for the State of Minas Gerais, Universidade Federal de Minas Gerais, Belo Horizonte 31630-901, Brazil; jacqueliness@ufmg.br; 6Department of Statistics, Institute of Exact Sciences, Universidade Federal de Minas Gerais, Avenida Antonio Carlos, Belo Horizonte 31270901, Brazil; thiagords@ufmg.br; 7Discipline of Population Oral Health, School of Dentistry, The University of Sydney, Sydney, NSW 2006, Australia; woosung.sohn@sydney.edu.au; 8Department of Operative Dentistry, Universidade Federal de Minas Gerais, Avenida Antonio Carlos, Belo Horizonte 31270901, Brazil; liasc@ufmg.br; 9Department of Community and Preventive Dentistry, Universidade Federal de Minas Gerais, Avenida Antonio Carlos, Belo Horizonte 31270901, Brazil

**Keywords:** analgesics, anti-inflammatory agents, non-steroidal, prescriptions, dentistry

## Abstract

To describe trends of dentist-prescribed non-steroidal anti-inflammatory drugs (NSAIDs) and analgesics, from January 2011 to December 2021, as well as to examine the relationship between these trends and characteristics of public oral health services in Minas Gerais, Brazil. In this time-series analysis, all drugs were classified according to the Anatomical Therapeutic Chemical classification system. Drugs categorized as NSAIDs (M01A), and other analgesics and antipyretics (N02B) were included for analysis. The outcome was the number of Defined Daily Doses (DDDs)/1000 inhabitants/year for NSAIDs and analgesics in each town. Covariates referred to characteristics of public oral health services, such as coverage, estimates of dental procedures, and frequency of toothache. Linear time-series regression models were used to determine the influence of covariates on the outcome. Overall, there were 58,482 prescriptions of NSAIDs recorded in thirty-eight towns, while 47,499 prescriptions of analgesics in forty-three towns. For each year, there was a 0.38 (*p* < 0.001), and 0.28 (*p* < 0.001) increase in the average log of DDD/1000 inhabitants/year for NSAIDs and analgesics, respectively. A positive association was detected between toothache (*p* < 0.001) and the prescription of NSAIDs. Over the eleven years, there was a general rising trend in the prescriptions. Toothache was the only characteristic of public oral health services associated with the prescription rates of NSAIDs, implying that as the frequency of toothaches increase, so do the prescriptions of NSAIDs in the studied towns.

## 1. Introduction

Dental practitioners play a vital role in the effective management of oral pain. When patients present with acute dental pain caused by issues such as dental caries, trauma, or dental procedures, practitioners should conduct a comprehensive examination. After determining the underlying causes and establishing a diagnosis, a tailored treatment plan aimed at alleviating the patient’s pain should be formulated. In certain situations, pharmacotherapy might be involved [1].

The medications frequently prescribed for acute oral and dental pain include non-steroidal anti-inflammatory drugs (NSAIDs), analgesics, and opioids. NSAIDs, such as ibuprofen or nimesulide, are often the first choice for managing oral pain due to their anti-inflammatory and analgesic properties [2,3]. Studies have shown that, compared to opioids, NSAIDs demonstrated great pain relief after dental extraction and postoperative endodontic pain [4,5,6]. In cases of severe pain, or where NSAIDs are contraindicated, paracetamol or metamizole, associated with codeine, might be an alternative therapeutic option [3]. Some studies indicated an increasing trend that dental practitioners are prescribing these drugs more frequently [7,8,9,10,11].

Park et al. assessed the longitudinal pattern of the top twenty most prescribed drugs by Australian dental practitioners from 2006 to 2018 [10]. Over a decade, a significant annual increase in the prescription count was reported for paracetamol + codeine, ibuprofen, and naproxen [10]. In Croatia, prescriptions of NSAIDs increased by approximately 46% between 2014 to 2018 [9]. The utilization of ibuprofen, as the preferred NSAID among German dentists, witnessed a notable surge, with the prescription rate rising from 61.9% in 2012 to 88.1% in 2016 [12]. Ibuprofen was also the most prescribed NSAID in Nigeria, Colombia, and Brazil [11,13,14]. The upward trend in the use of drugs in dentistry reflects improved healthcare accessibility for diverse populations, thereby indicating progress in public health. Nonetheless, inequalities related to access to pain medication have been reported [14,15,16].

Previous literature has demonstrated a positive association between the density of dentists, the percentage of the population accessing dental care within the past year, and household income with higher regional consumption rates of pain medication [15,16]. In a cross-sectional survey conducted in the southeastern region of Brazil, it was observed that prescriptions of NSAIDs and analgesics were noticeably lower in cities with limited access to dental healthcare services and few oral health teams [14]. However, these associations have not been evaluated over the long run.

Drug utilization research offers valuable insights regarding medication use within populations and the temporal fluctuations. Moreover, such research holds pivotal importance in guiding the allocation of healthcare resources and ensuring equitable access to drugs [17]. Nevertheless, the scarce and fragmented data on drug utilization in numerous developing nations, including Brazil, restricts our comprehension of the population consumption of drugs [17,18]. In this context, the objectives of this study were to describe trends of dentist-prescribed NSAIDs, and analgesics distributed through the Brazilian National Health System (BNHS). Additionally, we aimed to examine the relationship between these trends and the characteristics of public oral health services in Minas Gerais, Brazil.

## 2. Materials and Methods

### 2.1. Ethics Statment

The methods employed in this investigation are in accordance with the principles outlined in the Declaration of Helsinki. Prior to conducting the research, it underwent appreciation of the Research Ethics Board of the Universidade Federal de Minas Gerais on the 8 June 2018, resulting in a protocol (number: 88465118.8.0000.5149) and subsequent approval (number: 2.701.715).

### 2.2. Study Design and Data Source

This was a time-series analysis utilizing secondary data obtained from pharmaceutical claims of NSAIDs and analgesics prescribed by dental practitioners between 1 January 2011 and 31 December 2021. The investigation was conducted in the state of Minas Gerais, Brazil, known as the second most populous state in the country, with a population of approximately 20.5 million people. Notably, the state encompasses over 800 towns, characterized by great social disparities among them [19].

The BNHS provides comprehensive healthcare coverage to individuals of all age groups within the country. This coverage is accessible to the population without any direct costs at the point of delivery. The constitution guarantees the universal right to comprehensive care, including dental, pharmaceutical, and medical benefits of varying degrees of complexity. Recent data from the 2019 National Health Survey revealed that a significant portion of the population, specifically 71.5%, exclusively relied on the BNHS as their primary source of healthcare [20].

To investigate the prescribing trends in Minas Gerais, we examined secondary data from the Pharmaceutical Management System, known as Sigaf, in Portuguese. This software has been in use since 2010 and facilitates the pharmaceutical management of public pharmacies. It enables inventory registration, helping services track their stock of drugs and supplies. Additionally, it maintains a dataset with the history of drug dispensing, including the quantity and dosage prescribed to each patient, as well the prescriber’s name and license number [21]. Compliance with Sigaf is not compulsory; as such, not all towns in Minas Gerais use the tool.

The first stage of our research involved identifying prescriptions written by dental practitioners. A two-step cross-matching verification was performed for all prescriptions recorded by Sigaf. This process involved comparing the names and license numbers of the prescribers registered in Sigaf with the corresponding details found in the electronic files of the Regional Council of Dentistry, which maintains up-to-date records of all dentists licensed in the state. This verification was carried out using algorithms built with Python (v. 3.11.4) (Python Software Foundation, Wilmington, DE, USA). Then, to ensure that there were no inconsistencies (such as homonymous practitioners), a researcher (AJSC) carefully inspected all records in the dataset. As a result, the final dataset only included dental prescription records. A flowchart describing further details of the selection processes is shown in Figure 1.

For the subsequent phase of the study, all drugs prescribed by dental practitioners and recorded by Sigaf were classified following the guidelines for Anatomical Therapeutic Chemical (ATC) classification and Defined Daily Doses (DDDs) assignment, 2023 [22]. In the ATC system, drugs are categorized based on the physiological system or organ where they act [23,24]. Our analysis included drugs from the following ATC Classification System subgroups: M01A-anti-inflammatory and antirheumatic products, non-steroids; and N02B-other analgesics and antipyretics. Subsequently, the DDD for each prescribed drug was determined. The DDD corresponds to the average maintenance dose per day for a drug used for its main indication in adults [23,24]. We estimated the total number of DDD and DDD/1000 inhabitants/year for each town in Minas Gerais for both NSAIDs and analgesics between 2011 and 2021.

Our study followed the Reporting of Studies Conducted Using Observational Routinely Collected Health Data Statement for Pharmacoepidemiology (RECORD-PE) [25].

### 2.3. Variables

The outcome variable was DDD/1000 inhabitants/year for NSAIDs and analgesics. Covariates referred to characteristics of public oral health services, such as coverage (percentage of the population coverage by primary oral health care), estimates of dental procedures (first dental appointment/1000 inhabitants/year; extractions/1000 inhabitants/year; endodontics/1000 inhabitants/year), and toothache (toothache/1000 inhabitants/year). All covariates are quantitative and were sourced from official Brazilian government databases. Furthermore, all measurements were recorded at the municipal level. The data used in this study were collected during the specified research time frame from 2011 to 2021.

### 2.4. Statistical Analysis

Initially, we conducted descriptive statistical analysis to evaluate the outcome and covariates, including the calculation of proportions, central tendency, and variability. For the time-series data analysis, we employed a multiple linear regression model. It has been stated before that Sigaf does not store data from all towns in Minas Gerais. Therefore, in order to evaluate time trends, we have only considered data from towns that recorded prescriptions over the 11-year period under study.

The multiple linear time-series regression model is used to explain the relationship between two or more variables. It assumes a linear relationship between the covariate time series (*X*) and the outcome time series (*Y*) and aims to find the best-fitting line that describes the relationship. It is defined using the equation [26]:$$Y_{t}=X_{t}\beta +\phi_{p}\varepsilon_{t-p}+\cdots +\phi_{1}\varepsilon_{t-1}+\varepsilon_{t},\;\varepsilon_{t}\sim N\left(0,\;\alpha^{2}\right),\;\mathrm{for}\;t=1,\ldots ,n.$$

In the given model, the outcome time series is denoted as $Y_{t}$. The covariate time series matrix is denoted as $X_{t}$. The associated error is represented by $\varepsilon_{t\;}$, and the regression parameters are represented by the vector β. The coefficients $\phi_{1},\ldots ,\phi_{p}$ are associated with the lagged error terms to induce a dynamic structure into the model. It is assumed that the model errors have a mean of zero, are not correlated, and have a constant variance. In order to stabilize the variance and bring the data closer to a probability normal distribution, a logarithmic transformation was performed on the outcome. To verify if the coefficients of the covariates in the regression model were significant, a Student’s *t* test was used, with *p* < 0.05 considered as statistically significant. Residual standard error and degrees of freedom, as well as multiple adjusted R-squared, were indicated to evaluate the goodness-of-fit of the model. Missing data for the five covariates were imputed by calculating the arithmetic mean of the two closest available values [27]. Data analysis was performed using R version 4.3.1 (R Core Team, Vienna, Austria).

## 3. Results

### 3.1. NSAIDs

A total of thirty-eight towns recorded dental prescriptions of NSAIDs (n = 58,482) throughout the duration of the research. In these towns, there was a substantial rise of 230.48% in the count of NSAID prescriptions. The number of DDD rose from 14,908.17 in 2011 to 54,678.50 in 2021. The most frequently prescribed NSAIDs were ibuprofen (n = 34,395; 58.81%), nimesulide (n = 16,437; 28.11%), and diclofenac (n = 7595; 12.99%), which together accounted for nearly 100% of all NSAIDs. While ibuprofen was the most prescribed drug, its proportion among all NSAIDs exhibited temporal variation, reaching a peak of 76.99% (n = 5910) by 2021. In contrast, diclofenac showed a cumulative decline of −71.33% (Table 1).

The average DDD/1000 inhabitants/year for NSAIDs was 124.04 (S.D. = 134.92). The summary statistics for covariates can be found in Table 2.

The linear time-series regression model showed that the intercept was significant (coefficient 1.95; standard error 0.57; *p* < 0.001). In general, the effect of time was positive and significant at a 5% level. For each year, there was a 0.38 (standard error 0.08; *p* < 0.001) increase in the average log of DDD/1000 inhabitants/year. The dynamic components of residuals at 2, 3, and 4 years were significant and negative, showing a quality fit of the model (Table 3). The time-series regression model had an explanatory power of 82% (R^2^-Adj = 0.82). The individual intercept and trend for each town are presented in Appendix A.

After incorporating the covariates in the multiple linear time-series regression model, only toothache/1000 inhabitants/year remained significant (coefficient 0.00; standard error 0.00; *p* = 0.01) in the final model. This implies that as the frequency of toothache increases, prescriptions of NSAIDs also increase (R^2^-Adj = 0.80) (Table 4). The full and final multiple linear time-series model for each town are presented in Appendix A. The residual analysis was performed, and no evidence of violation of the model’s assumptions was observed.

### 3.2. Analgesics

In total, forty-three towns registered dental prescriptions of analgesics (n = 47,499). Within these towns, metamizole sodium (n = 31,557; 66.44%), and Paracetamol (n = 14,586; 30.71%) corresponded to 97.15% of all analgesics between 2011 and 2021. Except for the year 2011, metamizole was the most frequently prescribed analgesic over time. The number of DDDs for all pain relievers increased from 4472.00 in 2011 to 21,099.00 in 2021 representing a surge of 274.18% (Table 5).

The average DDD/1000 inhabitants/year for analgesics was 37.84 (S.D. = 55.26). Summary statistics for covariates can be found in Table 6.

The linear time-series regression model demonstrated that the intercept was not significant (coefficient 0.46; standard error 0.52; *p* = 0.38). In general, the effect of time was positive. For each year, there was a 0.28 (standard error 0.08; *p* < 0.001) increase in the average log of DDD/1000 inhabitants/year. The dynamic components of residuals at 2, 3, and 4 years were significant and negative, showing the quality fit of the model (Table 7). The time-series regression model had an explanatory power of 84.00% (R^2^-Adj = 0.84). Individual intercept and trend for each town are presented in Appendix A. Again, no evidence of the violation of the model’s assumptions was observed.

In the full multiple linear time-series regression model, none of the covariates were significantly associated with the outcome. Therefore, no covariates were included in the final model (Table 8). The full multiple linear time-series model for each town is presented in Appendix A.

## 4. Discussion

The analysis, utilizing data from Sigaf, revealed an overall increase in both the prescription count and DDD/1000 inhabitants/year of NSAIDs and analgesics prescribed by dental practitioners and dispensed under the BNHS. Ibuprofen and metamizole were the most frequently prescribed NSAID and analgesic, respectively. At the town-level, a positive association was observed between toothache prevalence and prescriptions of NSAIDs, whereas none of the covariates influenced the rates of analgesic prescriptions. This finding suggests that individual-level factors, such as clinical diagnosis and patient’s pain threshold, may play a significant role in prescribing decisions.

The preference of ibuprofen, over other NSAIDs observed in this study aligns with findings reported elsewhere [7,9,11,13]. This favorability might stem from ibuprofen’s well-established efficacy, safety profile and low cost. Current evidence demonstrates that, when compared to other NSAIDs, ibuprofen poses a relatively low risk of cardiovascular adverse effects, gastrointestinal bleeding, and renal impairment [28,29]. On the other hand, in Minas Gerais, diclofenac prescriptions have been decreasing over time. Similarly, research from Croatia showed a 22% decrease in diclofenac prescriptions [9], and in Australia there has been a decline in diclofenac consumption [10]. These trends could be attributed to diclofenac’s higher propensity for inducing adverse events, such as myocardial infarction, stroke, and hepatoxicity. Even with short-term use, diclofenac is recognized as the NSAID most likely to cause severe side effects [30,31]. The decrease in diclofenac usage might indicate that dentists are relying on evidence-based information when prescribing. Regardless, it remains crucial to exercise caution when prescribing NSAIDs due to their potential side effects and interactions with other medications. Providers should carefully consider each patient’s medical history, allergies, and any contraindications before prescribing NSAIDs.

Metamizole sodium was the most commonly prescribed analgesic by dental care providers in the towns in Minas Gerais. A nationwide survey, representative of Brazilian households, found that metamizole accounted for 52.8% of the primary drug used for pain relief by the population [32]. Although some studies suggest that metamizole is a safe choice when compared to other analgesics, such as opioids, it has been banned in several countries due to concerns regarding the risk of agranulocytosis [33,34]. In Brazil, metamizole can be purchased both with a prescription and over-the-counter medication. Given the restricted availability of metamizole in many parts of the world, direct comparisons with our data are limited.

The observed growth in NSAID and analgesic prescriptions in our study aligns with international trends of increasing drug utilization to treat painful dental inflammatory conditions. In Australia, Hollingworth et al. examined the patterns of drugs, listed on the Pharmaceutical Benefits Schedule, prescribed to dental patients during the period spanning from 2001 to 2012. Over these 12 years, NSAID prescriptions surged from 4039 in 2001 to 6855 in 2012, marking a nearly 70% increase [7]. Meanwhile, in Croatia, prescriptions of NSAIDs and analgesics by dental care providers soared between 2014 and 2018, leading to an overall increase of 46% over the five years [9]. In contrast, Halling et al. reported a decline of 100,000 (3.4%) painkiller prescriptions in Germany from 2012 to 2016 [12]. These variations among countries may be attributed to various factors, including the distinctive characteristics and scope of public healthcare systems, pain medication availability, and the range of drugs subsidized by governments, whether provided free or requiring copayments.

Dentists prescribe analgesic medications to mitigate the discomfort arising from untreated dental conditions, such as dental caries, periodontal diseases, and oral trauma, as well as pre- and post-operative interventions. Previous research showed a positive association between toothache and the prescription and consumption of pain relievers [35,36]. Moeller et al. reported that 10% of residents in British Columbia, Canada, experienced toothache during the study investigated period. Of these, nearly 80% used some painkillers [35]. Another cross-sectional survey conducted in Iran showed that pain relievers were commonly taken among those with dental pain [36]. Our findings provided additional evidence to support the positive association between the occurrence of toothache and the prescription of NSAIDs. Given the proven effectiveness of NSAIDs in managing various painful inflammatory dental conditions [2], it is logical for this class of medication to be a preferred choice among dentists. Interestingly, our study did not detect a significant role of toothache in the prescription of analgesics in the studied towns. These findings suggest the presence of unidentified factors that could potentially influence the prescribing patterns of pain relievers in Minas Gerais.

Other covariates, including population coverage by primary oral health care and estimates of dental procedures, did not exhibit a significant influence on the outcome. These covariates serve as proxy indicators for access to services. A higher number of dental procedures in certain regions suggests improved accessibility to curative dental care. While some cross-sectional studies have indicated a positive association between access to care and increased drug prescriptions [14,37], our time-series data analysis did not confirm this relationship with statistical significance. This suggests that additional individual-level factors, such as pain thresholds and clinical diagnoses, may play a central role in influencing dentists’ decisions to prescribe pain relievers [38,39]. Further investigations incorporating different covariates are necessary to comprehend the complex interactions pertaining to prescription patterns in dental practice.

This study is subject to inherent limitations commonly encountered in investigations utilizing secondary data, such as the inability to control and ensure data quality. The analysis was restricted to data from a limited number of towns (thirty-eight and forty-three) in the state of Minas Gerais, where NSAIDs and analgesics were prescribed over an eleven-year period. This limitation affects the external validity of the findings. Nonetheless, Sigaf remains a valuable source of information for research, especially given the current scarce and fragmented evidence about the populational usage of drugs in the Latin American region [17,18]. The Sigaf database, however, lacks information on specific diagnoses associated with drug prescriptions, making it impossible to evaluate the appropriateness and quality of the drug usage. Despite these drawbacks, to the best of our knowledge, this study represents the first time-series analysis to investigate prescribing patterns and associated factors regarding pain relievers in Brazil. The findings of this study offer insights about drug utilization, which may be particularly valuable for pharmaco-surveillance and can inform the development of strategies aiming to improve equitable and rational access to medication.

Future research should continue monitoring the rise in NSAID and analgesic prescriptions in dental practices, while conducting comparative analyses across diverse geographic regions. It is crucial to explore the factors influencing prescription patterns with a focus on individual-level factors. Furthermore, a comprehensive assessment of the appropriateness of prescribed NSAIDs and analgesics, dosage regimens, and potential interactions with concurrently administered medication is necessary. Addressing these gaps will help enhance the pharmaceutical management of public services and ensure patient safety related to drug utilization.

## 5. Conclusions

Over the eleven years, there was a noticeable upward trend in the prescription rates of the analyzed drugs, with ibuprofen and metamizole sodium as the most frequently prescribed NSAID and analgesic, respectively. Toothache was the only characteristic of public oral health services associated with the prescription rates of NSAIDs; as the frequency of toothache increased, so did NSAID prescription in the studied towns. This finding underscores the need for further research to identify potential underlying factors influencing drug usage within dental practice in Brazil.

## Figures and Tables

**Figure 1 ijerph-20-06795-f001:**
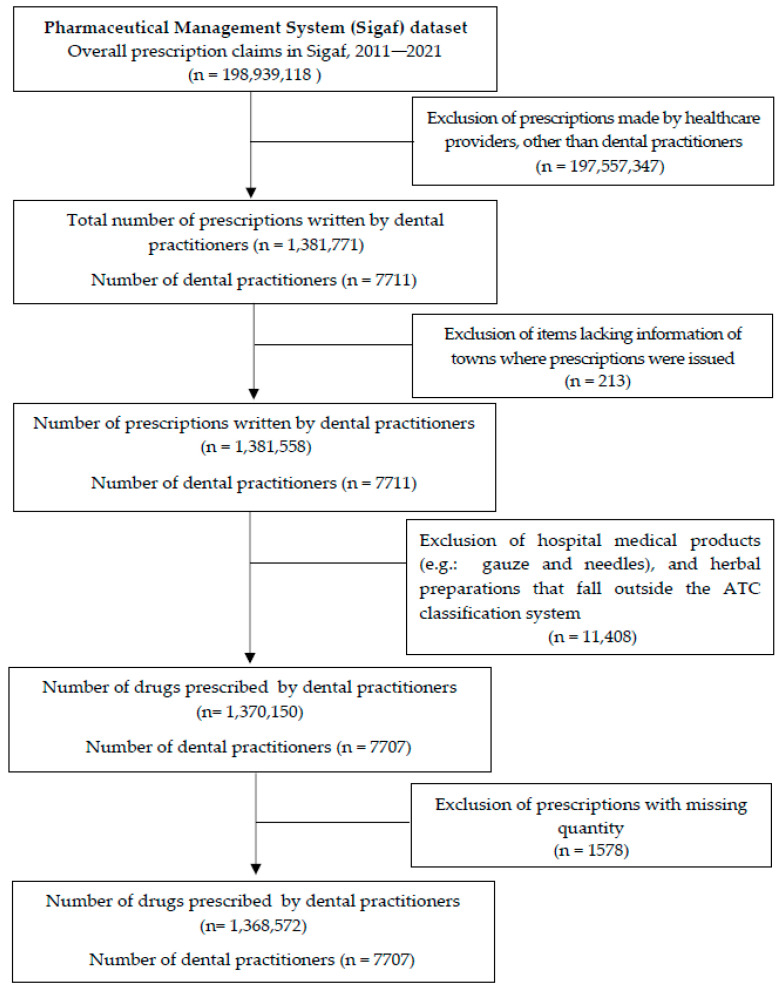
Flow chart of selection process.

**Table 1 ijerph-20-06795-t001:** Prescriptions count (n), percent of category (%), total defined daily dose, and accumulated annual percent change of non-steroidal anti-inflammatory drugs (NSAIDs), 2011–2021.

Drugs	2011			2016			2021			
	n	%	DDD	n	%	DDD	n	%	DDD	% Change
Diclofenac	708	30.48	5096.50	650	11.79	6034.75	203	2.64	1977.25	−71.33
Ibuprofen	1039	44.73	6578.17	3215	58.34	23,587.92	5910	76.99	43,747	468.82
Nimesulide	576	24.80	3233.50	1636	29.69	8678.50	1561	20.33	8926.25	171.01
Others	0	0	0	10	0.18	131.00	3	0.04	28	0
All NSAIDs	2323	100	14,908.17	5511	100	38,432.17	7677	100	54,678.50	230.48

**Table 2 ijerph-20-06795-t002:** Summary of the study outcome and covariates for NSAIDs, 2011–2021.

Variables	Mean (S.D.)	Minimum; Maximum
DDD/1000 inhabitants/year	124.04 (134.92)	0.02; 805.96
Percentage of the population coverage by primary oral health care	87.68 (23.24)	0.00; 100.00
First dental appointment/1000 inhabitants/year	333.97 (1081.77)	0.22; 9999.37
Extractions/1000 inhabitants/year	59.87 (46.31)	0.65; 332.55
Endodontics/1000 inhabitants/year	59.82 (48.02)	0.65; 340.90
Toothache/1000 inhabitants/year	36.94 (39.14)	0.01; 348.00

S.D.: Standard Deviation.

**Table 3 ijerph-20-06795-t003:** Linear time-series regression model for DDD/1000 inhabitants/year of NSAIDs.

	Coefficient	Standard Error	t-Value	*p*-Value
Intercept	1.95	0.57	3.40	<0.001
$e_{t-2}$	−0.21	0.05	−3.76	<0.001
$e_{t-3}$	−0.29	0.06	−5.17	<0.001
$e_{t-4}$	−0.33	0.06	−5.15	<0.001
Trend	0.38	0.08	4.50	<0.001

**Table 4 ijerph-20-06795-t004:** Full and final multiple linear time-series regression model for DDD/1000 inhabitants/year of NSAIDs and covariates.

	Full Coefficient	Standard Error	t-Value	*p*-Value	Final Coefficient	Standard Error	t-Value	*p*-Value
Percentage of the population coverage by primary oral health care	0.00	0.00	0.34	0.74				
First dental appointment/1000 inhabitants/year	−0.00	0.00	−0.15	0.88				
Extractions/1000 inhabitants/year	0.01	0.01	0.89	0.37				
Endodontics/1000 inhabitants/year	−0.00	0.00	−0.40	0.69				
Toothache/1000 inhabitants/year	0.00	0.00	2.20	0.03	0.00	0.00	2.35	0.01

**Table 5 ijerph-20-06795-t005:** Prescriptions count (n), percent of category (%), total defined daily dose, and accumulated annual percent change of analgesics from 2011 to 2021.

Drugs	2011			2016			2021			
	n	%	DDD	n	%	DDD	n	%	DDD	% Change
Metamizole Sodium	866	48.71	2127.00	2398	62.00	5759.05	4818	72.42	15,372.00	456.35
Paracetamol	908	51.07	2337.00	1422	36.76	3841.48	1242	18.67	3964.17	36.78
Others	4	0.22	8.00	48	1.24	60.33	593	8.91	1762.83	14,725.00
All analgesics	1778	100	4472.00	3868	100	9660.80	6653	100	21,099.00	274.18

**Table 6 ijerph-20-06795-t006:** Summary of the study outcome and covariates for analgesics, 2011–2021.

Variables	Mean (S.D.)	Minimum; Maximum
DDD/1000 inhabitants/year	37.84 (55.26)	0.01; 712.71
Percentage of the population coverage by primary oral health care	88.80 (21.88)	0.00; 100.00
First dental appointment/1000 inhabitants/year	331.88 (1071.30)	0.06; 11,102.26
Extractions/1000 inhabitants/year	106.43 (899.65)	0.18; 19,268.35
Endodontics/1000 inhabitants/year	106.99 (899.69)	0.65; 19,268.35
Toothache/1000 inhabitants/year	34.69 (37.46)	0.01; 348.00

S.D.: Standard Deviation.

**Table 7 ijerph-20-06795-t007:** Linear time-series regression model for DDD/1000 inhabitants/year of analgesics.

	Coefficient	Standard Error	t-Value	*p*-Value
Intercept	0.46	0.52	0.88	0.38
$e_{t-2}$	−0.27	0.05	−5.10	<0.001
$e_{t-3}$	−0.20	0.05	−3.74	<0.001
$e_{t-4}$	−0.29	0.06	−4.79	<0.001
Trend	0.28	0.08	3.68	<0.001

**Table 8 ijerph-20-06795-t008:** Full and final linear time-series regression model for DDD/1000 inhabitants/year of analgesics and covariates.

	Full Coefficient	Standard Error	t-Value	*p*-Value
Percentage of the population coverage by primary oral health care	0.00	0.00	0.48	0.63
First dental appointment/1000 inhabitants/year	0.00	0.00	0.75	0.45
Extractions/1000 inhabitants/year	0.01	0.01	0.81	0.42
Endodontics/1000 inhabitants/year	−0.01	0.01	−0.81	0.42
Toothache/1000 inhabitants/year	0.00	0.00	1.24	0.21

## Data Availability

The data that support the findings of this study are available from the corresponding author upon reasonable request.

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
