# Peer review of "Dental Pain Medication Prescriptions in Minas Gerais, Brazil (2011–2021): A Time-Series Analysis"

_ijerph, 2023, doi:10.3390/ijerph20186795_

Round 1

Reviewer 1 Report

“Prescription trends of pain medication for dental care in a Brazilian southeastern state from 2011 to 2021” was submitted to IJERPH.

This study aimed to describe trends in dentist-prescribed NSAIDs and analgesics and to examine the relationship between these trends and characteristics of public oral health services in an state of Brazil.

The authors concluded that there was a rising trend in the prescriptions, being toothache associated with prescribing rates of NSAIDs.

The manuscript deals with an interesting issue; however, there are several concerns related to the study.

Title: Please include the type of study.

Abstract

Line 30. The acronym ATC is not necessary. M01A and N02B must be described in an understandable form.

Line 33. “Data analysis was performed using R v.4.3.1” must be eliminated.

Lines 35-38. “standard error and coefficient” must be eliminated.

Considering that the objectives include "the relationship between these trends and characteristics of public oral health services", conclusions should be presented in this regard.

Methods

The type of study carried out must be presented.

The form to establish the characteristics of public oral health services is not detailed. What characteristics were intended to be evaluated?

Results

The characteristics of the public oral health services are not presented.

Discussion

Line 242. Please present possible explanations.

Lines 258-266. Compare these results with studies from other parts of the world.

Lines 289-292. Compare these results with previous studies.

Conclusions

Lines 332-333. “These increasing trends are somehow similar to prescribing patterns in dentistry globally”. Please delete. This information was presented in the discussion and is not adequate for the conclusions of this study.

As stated previously, considering that the objectives include "the relationship between these trends and characteristics of public oral health services", conclusions should be presented in this regard.

minor editing

Author Response

Comments: This study aimed to describe trends in dentist-prescribed NSAIDs and analgesics and to examine the relationship between these trends and characteristics of public oral health services in a state of Brazil. The authors concluded that there was a rising trend in the prescriptions, being toothache associated with prescribing rates of NSAIDs.

The manuscript deals with an interesting issue; however, there are several concerns related to the study.

Author’s response: Thank you for taking the time to review our manuscript. Below, we provide answers to each concern that was raised.

Comments: Title: Please include the type of study.

Author’s response: Thank you for your comment. The title has been amended.

Text revised: Dental Pain Medication Prescriptions in Minas Gerais, Brazil (2011-2021): A Time-Series Analysis.

ABSTRACT

Comments: Line 30. The acronym ATC is not necessary. M01A and N02B must be described in an understandable form.

Author’s response: Thanks for bringing the lack of clarity to our attention. We have revised the abstract as suggested.

Text revised: In this time-series analysis, all drugs were classified according to the Anatomical Therapeutic Chemical classification system. Drugs categorized as NSAIDs (M01A) and other analgesics and antipy-retics (N02B) were included for analysis.

Comments: Line 33. “Data analysis was performed using R v.4.3.1” must be eliminated.

Author’s response: The sentence was excluded from the abstract.

Comments: Lines 35-38. “standard error and coefficient” must be eliminated.

Author’s response: As suggested by the reviewer, the text has been edited accordingly.

Text revised: For each year, there was a 0.38 (p<0.001), and 0.28 (p<0.001) increase in the average log of DDD/1,000 inhabitants/year for NSAIDs and analgesics, respectively. A positive association was detected between toothache (p<0.001) and prescriptions of NSAIDs.

Comments: Considering that the objectives include “the relationship between these trends and characteristics of public oral health services”, conclusions should be presented in this regard.

Author’s response: This comment of the reviewer is very much pertinent. The abstract section has been amended as presented below.

Text revised: Over eleven-years under research, there was a general rising trend in the prescriptions. Toothache was the only characteristic of public oral health services associated with the prescription rates of NSAIDs, implying that as the frequency of toothache increases, so do the prescriptions of NSAIDs in the studied towns.

METHODS

Comments: The type of study carried out must be presented.

Author’s response: We apologize for not making this information clear in the manuscript. Our research was designed as a time-series analysis.

Text revised: This was a time-series analysis utilizing secondary data obtained from pharma-ceutical claims of NSAIDs and analgesics prescribed by dental practitioners between January 1, 2011, and December 31, 2021.

Comments: The form to establish the characteristics of public oral health services is not detailed. What characteristics were intended to be evaluated?

Author’s response: In our study, the characteristics of public oral health services included assessing the percentage of the population covered by primary oral health care, estimating dental procedures, and measuring the frequency of toothache. All of this data was collected from official Brazilian surveillance datasets

Text revised: Covariates referred to characteristics of public oral health services, such as, coverage (Percentage of the population coverage by primary oral health care), estimates of den-tal procedures (First dental appointment/1,000 inhabitants/year; Extractions/1,000 in-habitants/year; Endodontics/1,000 inhabitants/year), and toothache (Toothache/1,000 inhabitants/year). All covariates are quantitative and were sourced from official Bra-zilian government databases. Furthermore, all measurements were recorded at the municipal level. The data used in this study were collected during the specified re-search time frame from 2011 to 2021.

RESULTS

Comments: The characteristics of the public oral health services are not presented.

Author’s response: Once again, we apologize if this information was not clear in the main text. In fact, the characteristics of the public oral health services (covariates) are presented in Tables 2 and 6. The summary measures of those variables were presented for the group of towns, that registered prescriptions of NSAIDs (Table 2), and analgesics (Table 6).

Text revised: N/A.

DISCUSSION

Comments: Line 242. Please present possible explanations.

Author’s response: As suggested, some possible explanations were added to the main text.

Text revised:  This finding suggests that individual-level factors, such as clinical diagnosis and patient’s pain threshold, may play a significant role in prescribing decisions.

Comments: Lines 258-266. Compare these results with studies from other parts of the world.

Author’s response: As suggested, some discussion comparing international data were presented.

Text revised: On the other hand, in Minas Gerais, diclofenac prescriptions have been decreasing over time. Similarly, research from Croatia showed a 22% decrease in diclofenac prescriptions [9], and in Australia there has been a decline in diclofenac consumption [10]. These trends could be attributed to diclofenac’s higher propensity for inducing adverse events such as myocardial infarction, stroke and hepatoxicity. Even with short-term use, diclofenac is recognized as the NSAID most likely to cause severe side effects [30,31].

Comments: Lines 289-292. Compare these results with previous studies.

Author’s response: We appreciate the opportunity to improve the discussion of our manuscript. Some comparisons were included in the paragraph.

Text revised:  Moeller et al. reported that 10% of residents in British Columbia, Canada, experienced toothache during the study investigated period. Of these, nearly 80% used some pain-killers [35]. Another cross-sectional survey conducted in Iran showed that pain reliev-ers were commonly taken among those with dental pain [36].

CONCLUSIONS

Comments: Lines 332-333. “These increasing trends are somehow similar to prescribing patterns in dentistry globally”. Please delete. This information was presented in the discussion and is not adequate for the conclusions of this study.

Author’s response: The sentence was excluded from conclusions.

Text revise: N/A.

Comments: As stated previously, considering that the objectives include "the relationship between these trends and characteristics of public oral health services", conclusions should be presented in this regard.

Author’s response: We have edited this section to reflect the same content of the conclusion presented in the abstract.

Text revised:  Toothache was the only characteristic of public oral health services associated with the prescription rates of NSAIDs; as the frequency of toothache increased, so did NSAIDs prescription in the studied towns.

Reviewer 2 Report

This paper is an interesting study that analyzed the relationship between public oral health services and trends in the prescription of painkillers for dental care. However, please check the minor issues as follows:

In the introduction section, the need and purpose of the research are clear.  

The research methods section complies with research ethics, and the research design, data collection process, definitions of variables, and statistical analysis methods were appropriate.

- Please add the program used for statistical analysis.  

In the research results section, in Tables 2 and 6, please revise the (±) in ‘Mean (±)’ to standard deviation (S.D.), and write the full name of ‘S.D.’ at the bottom of the table.

 In the discussion section, a discussion of the study results, clinical significance, limitations of the study, and future research directions are well described.

Minor editing of English language required.

Author Response

Comments: This paper is an interesting study that analyzed the relationship between public oral health services and trends in the prescription of painkillers for dental care. However, please check the minor issues as follows:

In the introduction section, the need and purpose of the research are clear.

The research methods section complies with research ethics, and the research design, data collection process, definitions of variables, and statistical analysis methods were appropriate.

Author’s response: Dear reviewer, we greatly appreciate your acknowledgment of the methodological quality employed in our research.

Comments: Please add the program used for statistical analysis.

Author’s response: We apologize for any lack of clarity regarding this information in the main text. In order to conduct our analyses, we utilized R version 4.3.1 (R Core Team).

Text revised: Data analysis was performed using R version 4.3.1 (R Core Team).

Comments: In the research results section, in Tables 2 and 6, please revise the (±) in ‘Mean (±)’ to standard deviation (S.D.) and write the full name of ‘S.D.’ at the bottom of the table.

Author’s response: The text and tables have been corrected as per the reviewer's suggestions.

Text revised: 

Line 194: The average DDD/1,000 inhabitants/year for NSAIDs was 124.04 (S.D.=134.92).

Line 225: The average DDD/1,000 inhabitants/year for analgesics was 37.84 (S.D.=55.26).

Comments: In the discussion section, a discussion of the study results, clinical significance, limitations of the study, and future research directions are well described.

Author’s response: We appreciate the reviewer's recognition of the quality of the section.

Reviewer 3 Report

Dear Authors,

Attached are the minor reviews related to your article. It appears to be well described and, after minor reviews have been performed, ready for publication.

Moderate editing of English language required.

Author Response

Comments: Dear Authors,

I have been invited to review your work entitled “Prescription trends of pain medication for dental care in a Brazilian southeastern state from 2011 to 2021”. I believe it is a work of concern, however, there are minor issues that deserve revision for the acceptance of this work to the International Journal of Environmental Research and Public Health (MDPI).

Please, provide a point-by-point response, highlighting the corrections with a color mark

specific to each reviewer.

I have carefully revised the manuscript. I realize that I have indicated few corrections that

need to be made but this is given by the fact that the authors have accurately described their topic, methodologically present no errors or shortcomings that I can suggest change/adding, the conclusions are consistent and appropriate, as well as the references given. Figures and tables are correctly presented. The topic is interesting as it provides an overview updated to 2021 regarding the state of the art on analgesic prescribing in dentistry in Brazil. I believe there are no changes to be made other than those indicated below. This is from the methodological and presentation point of view of the manuscript.

Author’s response: Dear reviewer, thank you for acknowledging the merit of our study. As requested, we have addressed all of the reviewers' comments and concerns in this rebuttal letter.

Comments: Title

I suggest indicating the study design at the end of the title

Author’s response: Thank you for your comment. The title has been amended as presented below.

Text revised: Dental Pain Medication Prescriptions in Minas Gerais, Brazil (2011-2021): A Time-Series Analysis.

Comments: Keywords

I suggest that the keyword "dentistry" also be included.

Author’s response: The keyword dentistry, was added to the list.

Text revised: Keywords: analgesics; anti-inflammatory agents, non-steroidal; prescriptions; dentistry

Comments: Editorial issues

English editing by a native speaker is recommended, spelling, and editing errors should be corrected.

Author’s response: The manuscript was carefully and thoroughly proofread by Professor Sohn, who made all the necessary corrections for grammar and typographical errors.

Round 2

Reviewer 1 Report

The authors have clarified all concerns of the study.

minor